# HINER: Neural Representation for Hyperspectral Image

## ABSTRACT

As a prevalent scientific data format with extensive applications, the efficient compression of hyperspectral images (HSI) and ensuring high-quality downstream tasks have garnered significant attention. This paper introduces HINER, a novel approach for compressing HSI using Neural Representation. HINER fully exploits inter-spectral correlations by explicitly encoding of spectral wavelengths and achieves a compact representation of the input HSI sample through joint optimization with a learnable decoder. By additionally incorporating the Content Angle Mapper with the L1 loss, we can supervise the global and local information within each spectral band, thereby enhancing the overall reconstruction quality. For downstream classification on compressed HSI, we theoretically demonstrate the task accuracy is not only related to the classification loss but also to the reconstruction fidelity through a first-order expansion of the accuracy degradation, and accordingly adapt the reconstruction by introducing Adaptive Spectral Weighting. Owing to the inherent capability of HINER to implicitly reconstruct spectral bands using input wavelengths, it can generate arbitrary continuous spectra, even those absent in the original input. Consequently, we propose utilizing Implicit Spectral Interpolation for data augmentation during classification model training, thereby improving overall task accuracy on compressed data. Experimental results on various HSI datasets demonstrate the superior compression performance of our HINER compared to the existing learned methods and also the traditional codecs. Our model is lightweight and computationally efficient, which maintains high accuracy for downstream classification task even on decoded HSIs at high compression ratios.

## CCS CONCEPTS

• **Computing methodologies → Image compression**; **Hyperspectral imaging**; *Image representations.*

## KEYWORDS

Hyperspectral image compression, implicit neural representation, spectral embedding, classification on compressed HSI

## 1 INTRODUCTION

The hyperspectral image (HSI) uses tens of spectral bands across a wide range of electromagnetic wavelengths at each pixel position to capture the physical scene [1], by which it promises exceptional capabilities for tasks like object detection, material inspection, and

*ACM MM, 2024, Melbourne, Australia*
© 2024 Copyright held by the owner/author(s). Publication rights licensed to ACM.
ACM ISBN 978-x-xxxx-xxxx-x/YY/MM
https://doi.org/10.1145/nnnnnnn.nnnnnnn

scene analysis for applications in agriculture [2], aerospace industry [3], remote sensing [4], etc. However, compared with the three-channel RGB image, orders of magnitude more spectral channels in each HSI sample present practical challenges for storage and transmission, largely impeding its use in various applications. As a result, efficient lossy HSI compression is highly desired.

In addition to traditional rules-based HSI compression methods using transform [5] or linear prediction [6], over the past few years, there has been a growing interest in leveraging deep learning techniques for HSI compression [7–11]. Through the powerful modeling capabilities of neural networks, learned HSI compression has demonstrated noticeable compression efficiency improvement. As one of them, implicit neural representations (INRs) have gained increasing popularity for representing natural signals with intricate characteristics. The fundamental concept behind INR is to represent a signal as a tailored neural network, thus, the compression of the input signal is translated into the compression of the neural model itself. Such INR methods significantly diminish the requisite for extensive training data given the high cost of acquiring large-volume HSIs [12], and also streamline the decoding process. Zhang et al. [12] and Rezasoltani et al. [13, 14] have pioneered the exploration of neural representation for HSI compression. Both of them directly adopt the architecture of SIREN [15], which employs a cascade of Multi-Layer Perceptions (MLP) with periodic activation functions, for pixel-by-pixel compression of the input HSIs. However, such a pixel-wise approach is built upon the assumption of spatial redundancy and represents HSIs through spatial position embedding, which disregards the strong correlation across spectral bands, leading to performance limitations. Even worse, signal distortion induced by those lossy compression methods notably deteriorates the accuracy of the downstream task (e.g., classification), which makes them extremely difficult to promote in applications.

A practical HSI compression solution pursues 1) high-efficiency R-D performance and lightweight decoding complexity and 2) a negligible accuracy drop using decoded HSI for the downstream task. In principle, each HSI collects a sequence of "frames" (spectral bands) at serial wavelengths, analogous to a video containing a sequence of frames at serial timestamps, which, however, differ fundamentally in inter-frame (spectral) correlations. Intuitively, the frame difference in a video mainly owes to the temporal motion, assuming consistent pixel intensity of objects across all timestamps. In contrast, such a "frame" difference in an HSI is due to reflectance variation at each pixel across spectral bands, typically assuming stationary scenes without temporal motion (see Fig. 1 and Fig. 3). Consequently, how to efficiently exploit correlations within and across spectral bands is crucial for improving compression performance and also benefiting downstream task on compressed HSI.

To this end, we propose HINER, a novel spectral-wise neural representation for HSI. The proposed HINER employs a positional encoding followed by an MLP to embed the spectral wavelengths of the input HSI sample explicitly. Such an explicit embedding is capable of effectively characterizing and exploiting cross-band

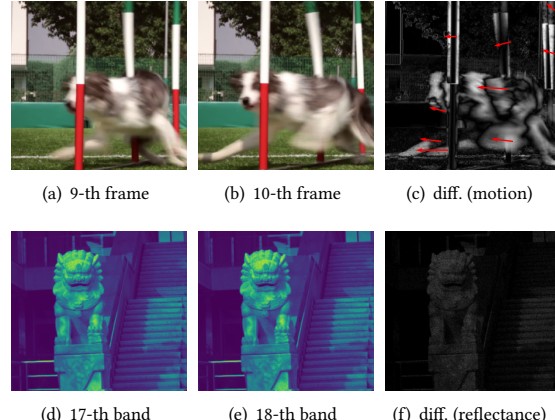

(a) 9-th frame     (b) 10-th frame     (c) diff. (motion)

(d) 17-th band     (e) 18-th band     (f) diff. (reflectance)

**Figure 1: Exemplified differences of the video frames (up, dog) and HSI bands (bottom, lion). Temporal motion leads to the difference of video frames while the difference of HSI bands owes to reflectance variation without motion.**

correlation, which is then fed into a learnable neural decoder to generate the corresponding decoded HSI. The pursuit of compression is achieved through collaboratively optimizing the encoder-decoder pair to generate more compact representations of spectral embedding and quantized decoder. Furthermore, we also propose to combine the Content Angle Mapper (CAM) measuring angle similarity between the reconstructed spectral band and its original counterpart and pixel-wise L1 loss, which contribute to maintaining global and local fidelity in signal reconstruction jointly.

Simultaneously, the impairment of downstream task performance on lossy compressed HSIs is a practical challenge [7]. Intuitively, the lossy compression may disrupt both the structural information and spectral continuity inherent in HSIs, which, without additional processing, will inevitably lead to accuracy degradation when optimized for vision tasks such as classification. To address this issue, we first employ a first-order Taylor expansion on the task accuracy degradation caused by the lossy compression, theoretically establishing an intrinsic connection between task accuracy and reconstruction fidelity. By introducing the Adaptive Spectral Weighting (ASW) network on the reconstructed HSIs with both task and reconstruction loss, the accuracy of downstream classification is greatly improved. Furthermore, owing to our INR-based compression approach, the monotonic mapping relationship between wavelengths and spectral bands enables the generation of arbitrary spectral bands, even those not present in the original HSI. Building upon this capability, we propose Implicit Spectral Interpolation (ISI) as a data augmentation technique for training classification model on compressed HSIs, resulting in significant enhancement in overall accuracy.

The main contributions of this paper are as follows:

(1) We propose HINER, a neural representation designed specifically for HSI. By introducing explicit encoding of wavelengths and global CAM loss, HINER effectively exploits spectral redundancy in HSI samples.

(2) We enhance the performance of downstream classification on lossy HSIs from two perspectives: adjusting the reconstruction to adapt to classification task through ASW, and improving the generalization of classification model with augmented data through ISI.

(3) Experimental results demonstrate the superior compression efficiency and comparable computational complexity of our proposed HINER compared to existing neural representation methods. Furthermore, there is a notable improvement in task accuracy when deploying classification on decoded HSIs at high compression ratios.

## 2 RELATED WORK

### 2.1 Implicit Neural Representation

Implicit Neural Representations (INRs) have gained widespread interest for its remarkable capability in representing diverse multimedia signals, including images [16, 17], videos [18–20], and neural radiance fields [21, 22]. Among them, NeRV [18] proposed the first frame-wise INR for video, which took frame indices as inputs to generate corresponding RGB frames through content-agnostic position encoding and a learnable decoder. Compared to previous pixel-wise INR methods (e.g., SIREN [15], Coin [23]), NeRV achieved better reconstruction quality while ensuring faster decoding. However, NeRV fully relied on the implicit learned decoder to characterize the input content and completely ignored the video content dynamics across frames. The subsequent HNeRV [24] proposed to explicitly embed frame content instead of index, leading to better reconstruction and faster model convergence for video sequence. Some recent works also attempted to capture temporal correlation by frame difference [25], optical flows [19], etc. On the other hand, INR has also attracted practitioners in HSI, including super resolution [26], reconstruction [27], fusion [28, 29], and compression [12–14], showing remarkable potential in practical applications.

### 2.2 HSI Compression

HSI compression [30–34] commonly employs transform coding [35] to convert HSI in the pixel domain to a latent space (e.g., frequency domain). Prominent transforms like Discrete Cosine Transform (DCT) [36] and Wavelet [37] utilized linear transformations that were generally comprised of a set of linear and orthogonal bases. However, such a linear transformation with fixed bases might not fully exploit the redundancy because the content of the underlying image block was non-stationary and did not strictly adhere to the Gaussian distribution assumption [38]. Chakrabart et al. [39] and Guo et al. [11] have demonstrated that real-world HSIs exhibited greater kurtosis and heavier tails than assumed Gaussian distribution, indicating a non-Gaussian nature of the HSI source. Therefore, devising transform to better exploit non-stationary content distribution is attracting intensive attention.

Over the past few years, learning-based HSI compression methods [9, 10, 40] have witnessed rapid growth. Dua et al. [9] and La et al. [10] firstly introduced Auto-Encoder (AE) for lossy HSI compression. Subsequently, Variational Auto-Encoders (VAEs) incorporating variational Bayesian theory, turned to represent latent features of input HSIs from a probabilistic perspective. Building

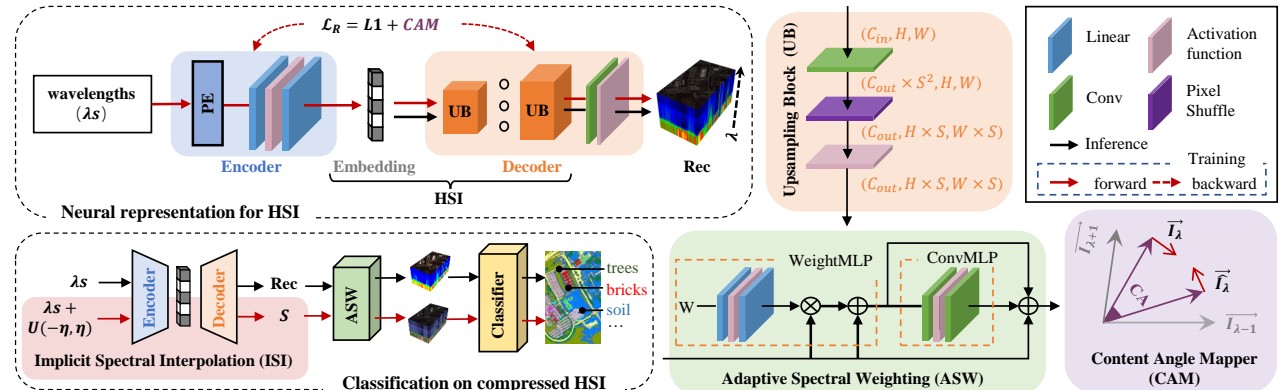

**Figure 2: The pipeline of our proposed HINER, the neural representation dedicated to compressing HSI, which also benefits downstream classification task on compressed HSI samples.**

upon the VAE architecture, Guo et al. [11] repurposed the hyperprior model [41] to compress HSI, where the student's T distribution [42] was used to replace the original Gaussian distribution, aligning more closely with the actual distribution of HSIs. Recently, Guo et al. [7] further introduced contrastive learning to preserve spectral attributes as much as possible for compression of HSI.

**INR-based HSI compression.** INRs provide a novel perspective on HSI compression by translating it into model compression. For instance, aforementioned Zhang et al. [12] and Rezasoltani et al. [13, 14] employed post-training quantization [43, 44] to compress models. However, even 16-bit quantization still resulted in significant performance loss (sometimes exceeding 1dB), which also indicated quite limited model capability of pixel-wise INRs with fully MLP-based network architecture. Considering HSIs can be treated as sequences akin to videos, well-established INRs for video compression can also be applied in HSI compression (though sub-optimal, as will be discussed in Sec. 3.1). Most video-based INRs follow a three-step compression pipeline: 1) pruning [45] to reduce model size; 2) quantization to reduce parameter bit-width; 3) entropy coding to reduce parameter statistical redundancy. Through these operations, the model is significantly compressed only with slight performance decline, attributed to elaborate network architecture enhancing model capability. For example, our proposed HINER subtly incorporates convolution, upsampling, GELU, etc., allowing the use of lower quantization bit-width (e.g., 8-bit) with a negligible reconstruction loss.

## 2.3 HSI Classification

HSI classification, which assigns each spatial pixel to a specific class based on its spectral characteristics, is the most vibrant field of research in the hyperspectral community and has drawn widespread attention [46]. Extracting more discriminative features is recognized as a crucial procedure for HSI classification [47], which achieves rapid advancements propelled by deep learning.

Many well-recognized networks have been widely and successfully applied in HSI classification task, including CNN [48–52],

AE [53], recurrent neural network (RNN) [54], graph convolutional network (GCN) [55]. Recently, transformer-based classification methods [56–58] show noticeable accuracy gains due to the self-attention mechanism, which effectively weights neighborhood information in dynamic input [59]. Hong et al. [56] developed a novel model called SpectralFormer (SF), capable of extracting features by aggregating multiple neighboring bands. Additionally, SF implemented cross-layer skip connections to reduce information loss during layer-wise propagation. Given that SF currently exhibits leading performance, we employ it as our baseline classification model for downstream task evaluation.

**Classification on compressed HSI.** Most current approaches in HSI classification continue to rely on uncompressed data due to the observed accuracy degradation induced by lossy compression. Unlike RGB images which can be visually appreciated by humans, compressed HSI will become completely useless if it cannot be applied to downstream tasks. The idea of using compressed images for classification dates back to the last century [60]. While some studies have explored the impact of lossy compression on HSI classification outcomes [61–65], primarily focusing on predicting classification accuracy for a given compressed HSI, our emphasis is on mitigating degradation for specific compressed samples without uncompressed ground truth.

## 3 METHOD

In this section, we begin by defining the optimization objective of neural representation for HSI (Sec.3.1). Subsequently, HINER, a neural representation for HSI compression, is proposed by exploiting the correlations within and across spectral bands (Sec. 3.2). Lastly, We theoretically analyze and address the issue of accuracy degradation in classification task on compressed HSI. (Sec. 3.3). An overview of our overall pipeline is illustrated in Fig. 2.

## 3.1 Preliminary

Let $I = \{I_\lambda\}_{\lambda=\alpha}^{\beta} \in \mathbb{R}^{H \times W \times C}$ denote an input HSI with spatial resolution of $H \times W$ and a total of $C$ spectral bands spanning the wavelength range $\lambda \in [\alpha, \beta]$. The objective of neural representation is to model a mapping function $\mathcal{F}$ from the embeddings $e$ to the HSI

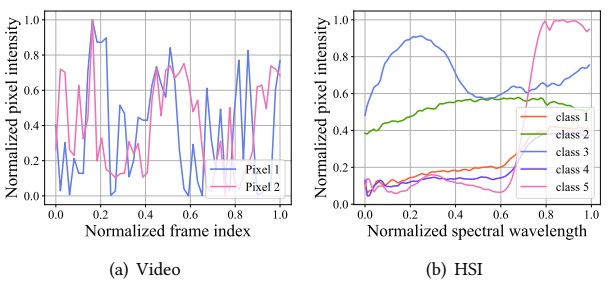

(a) Video

(b) HSI

**Figure 3: Pixel intensity distribution in fixed spatial position.**

$I: \mathcal{F}(e) \rightarrow I$ using a neural network. This work suggests to process HSI spectral-wisely. Given the spectral band $I_\lambda$ with wavelength $\lambda$, a learnable decoder $\mathcal{D}(\cdot)$ is employed for reconstruction by inputting spectral embedding $e_\lambda$. Our goal is to minimize the distortion between the input $I_\lambda$ and its reconstructed counterpart $\hat{I}_\lambda$ with the restricted model parameter $\theta$. As a result, the rate-distortion (R-D) optimization objective can be formulated as:

$$\arg\min \sum_{\lambda=\alpha}^{\beta} \mathcal{L}(I_\lambda, \hat{I}_\lambda) = \arg\min_{e, \mathcal{D}} \sum_{\lambda=\alpha}^{\beta} \mathcal{L}\left(I_\lambda, \mathcal{D}(e_\lambda)\right),$$
$$\text{s.t.} \quad \theta(e) + \theta(\mathcal{D}) \leqslant \theta, \tag{1}$$

where $\mathcal{L}$ represents the distortion loss. The bitrate $\theta(e)$ used for embeddings and the decoder parameter $\theta(\mathcal{D})$ collectively comprise the overall bitrate consumption, subject to the constraint of $\theta$.

As a comparative neural representation in video compression, NeRV [18] completely relied on a learnable decoder for implicit representation without any content embeddings. Since the embedding is generated by fixed position encoding of temporal indices, the only consumed bitrate is $\theta(\mathcal{D})$ without $\theta(e)$. HNeRV [24] firstly proposed the hybrid neural representation framework, which incorporated a learnable encoder to produce additional embeddings from frame content. Through a small amount of bitrate consumption by $\theta(e)$, such explicit content embeddings greatly improved the coding efficiency and model convergence.

Although explicit content embedding has demonstrated remarkable performance in video compression, an accompanying issue has arisen: can this success be replicated directly on HSI? As mentioned above, frame differences in a video primarily stem from non-monotonic temporal motion, which makes monotonic frame indices inadequate for capturing pixel correlations among neighboring frames (see Fig. 3(a)). Therefore, capturing differentiated content from each frame can yield better temporal embedding compared to content-agnostic frame indices. Conversely, in HSI, such "frame" differences originate from reflectance variation at each object across spectral wavelengths, where we often assume stationary objects without temporal motion. As can be seen in Fig. 3(b), there is a potential mapping relationship between wavelengths and pixel intensities (each pixel in HSI corresponds to a specific object class, such as tree, soil, etc.). Consequently, content embedding is sub-optimal for representing HSI which requires fully leveraging spectral correlation. To address this, we propose HINER, a neural representation fully exploiting spectral redundancy in Sec. 3.2.

Furthermore, we theoretically investigate and overcome the problem of performance degradation in downstream classification on compressed HSIs built upon the characteristic of HINER in Sec. 3.3.

## 3.2 Neural Representation for HSI

**Spectral Wavelength Embedding.** To capture spectral correlation, we take a straightforward yet highly effective approach by *explicitly embedding spectral wavelength* $\lambda$.

In the specific implementation, HINER explicitly encodes the normalized $\lambda \sim U(0, 1)$ using a learnable encoder $\mathcal{E}$ to generate the spectral embeddings $e = \{e_\lambda\}_{\lambda=\alpha}^{\beta}$, which is then forwarded to the decoder $\mathcal{D}$ for the reconstruction of the HSI. According to Eq. (1), the encoder does not consume bitrate and is only used to produce the spectral embeddings that need to be further encoded. However, considering that the training time of the entire neural representation model is equivalent to the encoding time of the HSI, it is crucial to design an efficient encoder with the following two characteristics: *1) Maintaining a low level of computational complexity; 2) Efficiently capturing spectral correlation.*

Inspired by the practice of [21] in neural radiance fields, $\mathcal{E}$ is built as a composition of two functions $\mathcal{E} = \mathcal{M} \circ \mathcal{P}$, by which

$$e_\lambda = \mathcal{E}(\lambda) = \mathcal{M}\left(\mathcal{P}(\lambda)\right), \tag{2}$$

where $\mathcal{M}$ stands for a tiny learnable MLP layer, and $\mathcal{P}$ denotes the frequency Positional Encoding (PE) [21, 66] to map $\lambda$ into a higher dimensional space $\mathcal{P} : \mathbb{R} \rightarrow \mathbb{R}^{2l}$, i.e.,

$$\mathcal{P}(\lambda) = \left(\sin(b^0 \pi \lambda), \cos(b^0 \pi \lambda), \ldots, \sin(b^{l-1} \pi \lambda), \cos(b^{l-1} \pi \lambda)\right). \tag{3}$$

The rationale behind not directly inputting $\lambda$s into the MLP layer without positional encoding is due to the well-known spectral bias [16, 67] in MLP. This bias tends to prioritize learning low-frequency components of the signal, potentially leading to the network's inability to adequately model high-frequency variation. [21, 66, 67]. This phenomenon is detailed in supplementary material.

By jointly optimizing the encoder and decoder, such a lightweight encoder is sufficient for information extraction and facilitates faster model convergence in training, which is also known as the encoding process for INR methods. The bitrate overhead of such spectral embedding is negligible (see Sec. 4.4), but it dramatically improves the performance of HINER by exploiting the inter-band correlation for better coding efficiency.

**Content Angle Mapper.** In general, INR models are typically optimized using L-p loss function [14, 24, 68]. However, the pixel-wise L-p loss lacks the ability to supervise global content similarity within spectral band. To address this limitation, some recent works [18, 69] introduce the Structure Similarity Index Measure (SSIM) loss by considering the correlations in luminance, contrast, and structure of the images, which may not apply to single spectral band. Drawing inspiration from the spectral angle mapper used for pixel correlation analysis [70, 71], we introduce the Content Angle Mapper (CAM) to calculate the angle between the original spectral band vector $\vec{I}_\lambda$ and its reconstructed counterpart $\vec{\hat{I}}_\lambda$. Minimizing CAM enables us to spatially exploit global content correlation in each spectral band. Additionally, L1 loss is also incorporated for pixel-wise supervision, which is proved to be more appropriate for scenes characterized by complex textures with high-frequency

information [25, 72]. Consequently, the optimization objective for training HINER can be formulated as:

$$\mathcal{L}_R = \underbrace{\sum_{\lambda=\alpha}^{\beta} ||\hat{I}_\lambda - I_\lambda||}_{L1 \ loss} + \gamma \cdot \underbrace{\sum_{\lambda=\alpha}^{\beta} \frac{180}{\pi} \arccos\left(\frac{\vec{\hat{I}}_\lambda^T \cdot \vec{I}_\lambda}{||\vec{\hat{I}}_\lambda^T||_2 ||\vec{I}_\lambda||_2}\right)}_{CAM}, \quad (4)$$

where the vector $\vec{I}_\lambda \in \mathbb{R}^{m \times 1}$ denotes the flattened spectral band $I_\lambda$, $m = H \times W$ is determined by spatial resolution, and $\gamma$ is introduced to make a trade-off between these two losses.

**Compression.** To further reduce the actual bitrate consumption of our HINER, we follow HNeRV and employ the same quantization and entropy coding methods for model compression. In model quantization, the floating-point vector $\mu_{float}$ (e.g., weight or bias in a convolutional layer) is quantized using:

$$\mu_{int} = clip\left(\lfloor \frac{\mu_{float} - \min(\mu)}{s_\mu} \rceil, 0, 2^b - 1\right),$$
$$\text{where} \quad s_\mu = \frac{\max(\mu) - \min(\mu)}{2^b - 1}, \quad (5)$$

$\lfloor \cdot \rceil$ rounds the input to the nearest integer. $b$ denotes the quantization bit-width, and $s_\mu$ is the linear scaling factor. We utilize the Huffman [73] coding as the lossless entropy coding method to further compress model parameters after quantization.

## 3.3 Classification on Compressed HSI

HSI classification refers to assigning a predefined label to each individual pixel [74], which is similar to the semantic segmentation task for the RGB image. The degradation in classification performance on compressed HSIs is related to the intrinsic characteristics of HSI, where pixel intensity corresponds to the spectral reflectance of objects across multiple spectral bands [7]. As depicted in Fig. 3, two categories with similar spectral reflectance (e.g., class 1 and class 4) may become indistinguishable after lossy compression. We enhance the performance of classification on lossy HSIs from two perspectives: 1) adjusting the compressed reconstruction to adapt to classification task; 2) improving the generalization of classification model with augmented data.

**Adaptive Spectral Weighting (ASW).** We first conduct theoretical analysis on the classification loss $\mathcal{L}_C$. Having the classification model parameterized by $\theta$, the original uncompressed HSI $I$, as well as the reconstructed HSI $\hat{I}$, we employ additive noise [75, 76] to model the compression loss, i.e., $I = \hat{I} + u(\hat{I})$. Consequently, the task performance degradation induced by compression can be defined as:

$$\mathbb{E}\left[\mathcal{L}_C(\theta, I) - \mathcal{L}_C(\theta, \hat{I})\right]. \quad (6)$$

Here, $\mathcal{L}_C(\theta, I)$ and $\mathcal{L}_C(\theta, \hat{I})$ stand for classification losses on uncompressed and compressed HSI respectively. With the first-order expansion on Eq. (6), we will derive:

$$\mathcal{L}_C(\theta, I) - \mathcal{L}_C(\theta, \hat{I}) \approx \nabla_{\hat{I}} \mathcal{L}_C(\theta, \hat{I})^T \cdot u(\hat{I}), \quad (7)$$

where the degradation is approximated as the first-order error for task loss $\mathcal{L}_C$. An intuitive solution is to optimize the input $\hat{I}$ based on the gradient $\nabla_{\hat{I}} \mathcal{L}_C(\theta, \hat{I})^T$ while ensuring $\hat{I}$ close to $I$, characterized by $||I - \hat{I}|| = ||u(\hat{I})|| < \epsilon$. However, directly

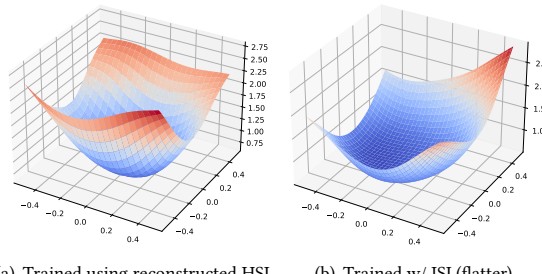

(a) Trained using reconstructed HSI     (b) Trained w/ ISI (flatter)

**Figure 4: Loss landscape [78] of trained classification model.**

optimizing $\hat{I}$ can be challenging to converge due to its high spatial and spectral resolution. Thus, we introduce a tiny learnable module named Adaptive Spectral Weighting (ASW) to adjust $\hat{I}$, as shown in Fig. 2. With ASW, the optimization of $\hat{I}$ is converted into the optimization of network parameters, which can be easily solved by gradient descent:

$$\theta_{ASW} = \arg\min \mathcal{L}_C + \beta \cdot \mathcal{L}_R, \quad (8)$$

where optimizing $\mathcal{L}_C$ aims to make the gradient $\nabla \mathcal{L}_C \to 0$ and optimizing $\mathcal{L}_R$ aims to constrain $u$.

ASW first spectral-wisely re-weights the reconstructed HSI by multiplying learned factors, followed by an MLP comprising 1x1 conv for cross-spectral information aggregation. The rationale behind ASW lies in the varying importance of spectral bands for reconstruction and downstream classification [77]. Thus, ASW facilitates the translation from perception-oriented reconstruction to classification-oriented reconstruction. More details can be found in the supplementary material.

**Implicit Spectral Interpolation (ISI).** HINER establishes a monotonic continuous mapping from spetral wavelengths to spectral bands. This enables HINER to reconstruct corresponding spectral bands for arbitrary wavelengths, even if these wavelengths or bands do not exist in the original discrete HSI sample (in some literatures, this function is also refered to as HSI reconstruction [79] or spectral super-resolution [80]). Leveraging this continuous mapping, we can easily construct an augmented sample set $\mathcal{S}$ containing multiple randomly sampled HSI by adding random variables to the input wavelengths of HINER

$$\mathcal{S} = \sum \mathcal{HINER}\left(\lambda + U(-\eta, \eta)\right), \quad (9)$$

where $U(-\eta, \eta)$ represent a uniform distribution that adds random variables to $\lambda$. Training the classification model with the augmented data set $\mathcal{S}$ significantly enhances performance on compressed HSI. It's important to note that we do not introduce ground truth HSI during training, thus improving the practical applicability of ISI. The rationale behind ISI lies in the fact that data augmentation can enhance the classification model's generalization on compressed HSI data [76, 81–83], thereby leading to improved accuracy. One intuitive manifestation of generalization is the flatness of the loss landscape [78]. A flatter loss landscape, indicative of better generalization, exhibits relatively small loss changes under parameter perturbations, whereas a sharp loss landscape indicates otherwise. As depicted in Fig. 4, the classification model trained with ISI exhibits

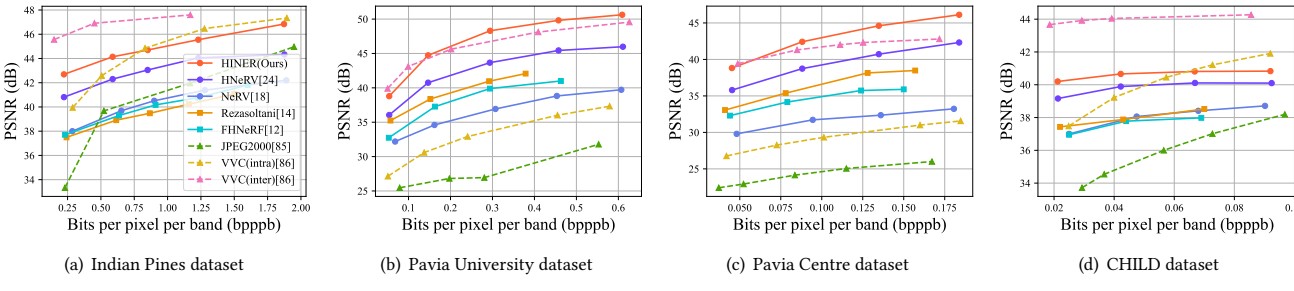

(a) Indian Pines dataset     (b) Pavia University dataset     (c) Pavia Centre dataset     (d) CHILD dataset

**Figure 5: R-D performance comparisons across HSI datasets.**

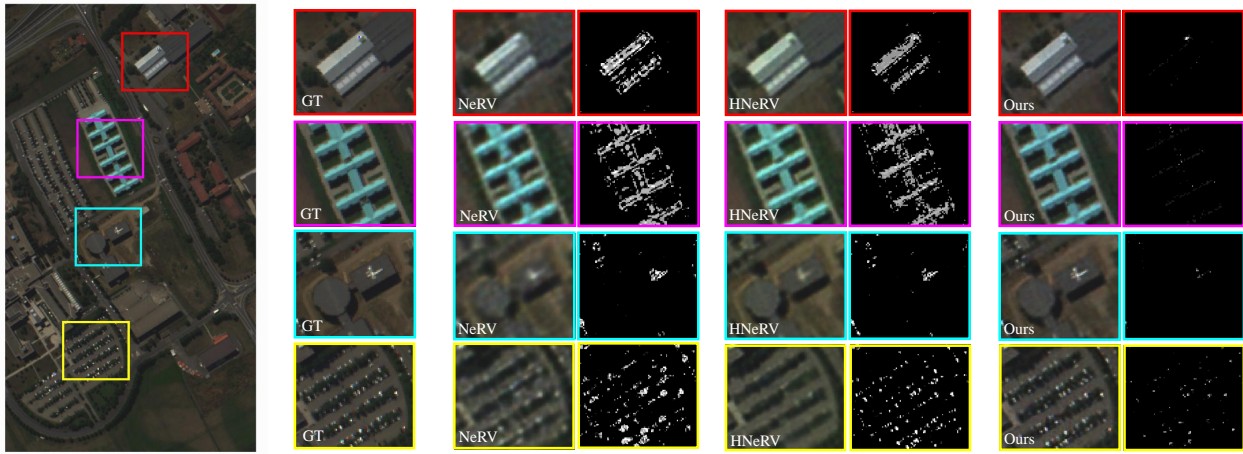

**Figure 6: Visualization comparisons of NeRV, HNeRV and our HINER on Pavia University dataset. The residual between the reconstruction and the ground truth (GT) is accompanied by each reconstructed result.**

a flatter loss landscape, in which Fig. 4(a) can also be considered a special case of $\eta = 0$. We provide detailed explanation in the supplementary material.

## 3.4 Discussion

So far, we have developed a neural representation framework, HINER, with a series of optimizations tailored for HSI, aligning better with its intrinsic characteristics and also aiding in maintaining performance of downstream tasks on compressed reconstruction. Unlike content-agnostic NeRV [18] or content-embedded HNeRV [24], we attempt to explicitly exploit cross-spectral correlations through wavelength embedding and fully supervise local and global reconstruction fidelity within a specific band by combining L1 and CAM loss. Our HINER also differs significantly from the previous pixel-wise compression methods [12, 14] which neglect spectral redundancies. We also thoroughly consider the degradation of downstream classification task caused by lossy compression. By using ASW for reweighting the compressed reconstruction and ISI for data augmentation in training classification model, we effectively address the issue of loss in task accuracy, which was not mentioned in previous works [12, 14].

## 4 EXPERIMENTS

### 4.1 Setup

**Datasets.** We conduct the evaluation on four popular HSI datasets with varying resolutions: 1) The Indian Pines dataset with size of $145 \times 145 \times 200$; 2) The *Pavia University* dataset with size of $610 \times 340 \times 103$; 3) The *Pavia Centre* dataset with size of $1096 \times 715 \times 102$; 4) The *CHILD* [84] dataset with size of $960 \times 1056 \times 145$.

**Metrics.** For compression performance comparison, *Peak Signal-to-Noise Ratio (PSNR)* is used to measure the reconstruction quality, and *bits per pixel per band (bpppb)* reports the consumption of compressed bitrate. The classification performance on compressed HSI is evaluated using *Overall Accuracy (OA), Average Accuracy (AA)*, and *Kappa Coefficient ($\kappa$)*.

**Baselines.** For compression performance comparison, we select the frame-wise neural representation methods, namely NeRV [18] and HNeRV [24], as well as the pixel-wise methods specifically designed for HSI compression, FHNeRF [12] and Rezasoltani [14]. JPEG2000 [85] and VVC [86] are exemplified to represent traditional image and video codecs. For JPEG2000, we employ OpenJPEG to independently compress each spectral band. For VVC, we utilize the reference software VTM 16.0 and conduct compression experiments

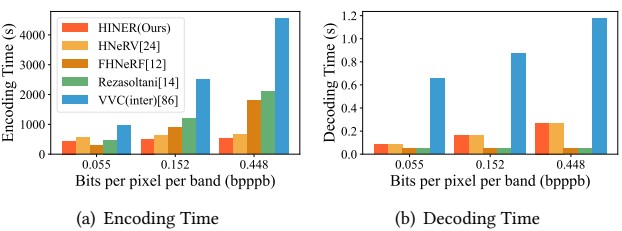

(a) Encoding Time    (b) Decoding Time

**Figure 7: Encoding & decoding complexity comparisons.**

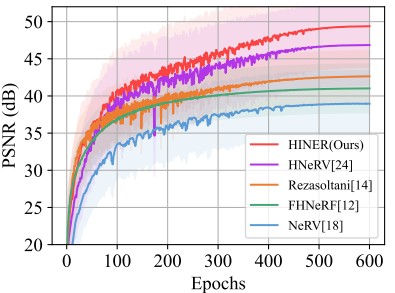

**Figure 8: Regression capacity comparisons.**

with both intra and inter (i.e., Random Access) profiles. The intra profile compresses each spectral band separately while the inter profile processes HSIs in spectral order as the coding of videos. For downstream classification, we use SpectralFormer (SF) [56] as the baseline, known for its leading performance.

**Implementation.** We faithfully reproduce the compared methods following their default settings on the HSI dataset. For the training of HINER, we adopt the Adam optimizer [87] with a batch size of 1. The initial learning rate is 0.001 with a cosine descent strategy. Stride sizes used for upsampling in the decoder are configured at (5, 3, 2, 2) for Indian Pines, (5, 4, 3, 2) for Pavia University, (5, 4, 3, 2, 2) for Pavia Centre, and (5, 4, 4, 2, 2) for CHILD, respectively. Unless specified otherwise, all experiments are conducted using PyTorch with an Nvidia RTX 3090 for totally 300 epochs. For classification, the learning rate is 0.0005 with 0.005 weight decay. Epochs for Indain Pines and Pavia University are 300 and 480, respectively.

## 4.2 HSI Compression

**Performance.** We present the R-D curves in Fig. 5 across various datasets. The proposed HINER clearly outperforms other neural representation methods. Leveraging efficient spectral wavelength embeddings, HINER not only significantly surpasses the content-agnostic method (i.e., NeRV), but also proves better than the content embedding method (i.e., HNeRV) in the HSI dataset. Compared with pixel-wise FHNeRF and Rezasoltani, such a band-wise representation of HINER exhibits superior rate-distortion advantage. Furthermore, our method is far better than the earlier image codec JPEG2000 across all datasets and superior to the VVC intra coding in Pavia University and Pavia Centre, in which our HINER is even comparable with the VVC inter coding. However, there is still a performance gap between learning-based methods and the VVC inter coding in Indian Pines and CHILD. One possible reason is that these two datasets capture simple scenarios with few texture information, which is easy for motion prediction in VVC thus greatly improving the coding efficiency. Fig. 6 visualizes the reconstruction results of neural representation methods on Pavia University. Notably, our method exhibits a much closer reconstruction to the original data.

**Complexity.** We also report the encoding and decoding time of our HINER, as well as HNeRV, FNeRF, Rezasoltani, and VVC inter profile in Fig. 7. Our model presents faster encoding and decoding compared to VVC, e.g., up to 10 × /6× encoding/decoding time reduction. The encoding of ours is also faster than HNeRV due to the lightweight spectral encoder. When compared with FHNeRF and Rezasoltani, spectral-wise HINER exhibits faster encoding, and

the gap further increases with model size. Although the inclusion of the upsampling and GELU [88] operation slows down our decoding than fully MLP-based FHNeRF and Rezasoltani, HINER achieves a PSNR improvement of more than 5 dB while still maintaining over 400 band-per-second decoding speed at Pavia University.

**Regression.** To evaluate the efficiency of HINER, we conduct comparisons regarding regression capacity in Fig. 8. As shown, those methods incorporating additional information embeddings, i.e., HNeRV and ours, lead to better reconstruction quality and faster model convergence speed than others. Moreover, our method shows the best performance, indicating the effectiveness of spectral embedding for HSI representation. An interesting observation is that Rezasoltani and FHNeRF exhibit rapid saturation in earlier training, probably due to pure MLP architecture limits the model capacity, and such pixel-wise representation is inadequate to capture cross-spectral redundancies.

## 4.3 Classification on Compressed HSI

When lossy compressed HSIs are widely used for storage and transmission due to their lower space-time resource occupation, downstream servers will not access the original ground truth data. Thus, related services, such as classification, have to rely solely on the compressed HSI. However, as shown in Table 1, lossy compression results in a significant degradation when optimized for classification (SF♣ vs. SF), primarily due to the spectral information that determines the class of objects has been compromised. Our proposed ASW and ISI, effectively alleviate degradation and maintain considerable accuracy even under high compression ratios, e.g., up to ×109, approaching the levels trained with ground truth. Notably, our approach even surpasses the performance of SF trained with ground truth data in the IndianPine. One potential explanation for this phenomenon is that our theoretical framework aids in mitigating data bias [89, 90] between the training and testing sets to some extent, when $u(I)$ in Eq. 7 is interpreted as a measure of data bias. This is an interesting topic for further study in the future.

## 4.4 Ablation Study

**Spectral Wavelength Embedding.** As mentioned above, by explicitly encoding the spectral wavelength $\lambda$s, the spectral correlation is embedded to assist the decoder reconstruction. One deduction is that when we randomly reorder the spectral bands, i.e., shuffling the original mapping from wavelengths to corresponding bands,

**Table 1: Quantitative performance of classification. ♣ represents the results trained with compressed HSI without ground truth. CR denotes compression ratio. The best results are highlighted in bold.**

| Datasets | Methods | CR | OA (%) | AA (%) | $\kappa$ |
|----------|---------|-----|--------|--------|----------|
| IndianP | SF | ×1 | 81.76 | 87.81 | 0.7919 |
| | SF ♣ | ×28 | 79.15 | 84.27 | 0.7633 |
| | Ours ♣ | | **87.03** | **90.99** | **0.8519** |
| PaviaU | SF | ×1 | 91.07 | 90.20 | 0.8805 |
| | SF ♣ | ×109 | 86.29 | 87.89 | 0.8203 |
| | Ours ♣ | | **88.93** | **88.96** | **0.8529** |

the permutation of spectra would be disturbed, thereby affecting the reconstruction results of HINER. The *case 1* in Table 2 confirms this deduction, in which our performance suffers from the shuffle operation compared to the default configuration. However, as a comparison, HNeRV is immune from the band shuffle without any performance loss, indicating that content embeddings fail to capture the inter-band correlation.

In addition, we also examine the effectiveness of the explicit encoder by *case 2* in Table 2. We solely remove the encoder $\mathcal{E}$ so that the input HSI is fully represented by the decoder with the fixed position encoding as in NeRV. As observed, such a pattern greatly decreases the coding efficiency, which illustrates the effectiveness of our explicit encoder in learning spectral wavelength embeddings for the decoder.

**Table 2: Ablations on spectral wavelength embedding.**

| | | w/o shuffle | w/ $\mathcal{E}$ | Indian | PaviaU |
|---|---|---|---|---|---|
| HNeRV | case 1 | ✗ | ✓ | 44.33 | 43.66 |
| | default | ✓ | ✓ | 44.33 | 43.66 |
| HINER | case 1 | ✗ | ✓ | 45.50 | 46.52 |
| | case 2 | ✓ | ✗ | 45.55 | 46.67 |
| | default | ✓ | ✓ | **46.03** | **47.17** |

**Embedding Size.** As mentioned in Sec. 3.1, the consumed bitrate of wavelength embeddings $\theta(e)$ and the decoder parameter $\theta(\mathcal{D})$ comprise the overall compressed bitrate. Given a certain rate constraint, it is necessary to make a trade-off between $\theta(e)$ and $\theta(\mathcal{D})$ for optimal compression efficiency. In Table 3, we evaluate the impact of embedding size on the reconstruction quality under a fixed total size of 0.5MB with 150 training epochs. We use size of $height \times width \times channel$ to denote a certain band embedding, thereby changing $\theta(e)$. It is suggested that the embedding size of $6 \times 3 \times 16$ with only 5% bitrate consumption is the optimal choice.

**Content Angle Mapper.** Table 4 presents a quantitative comparison under different loss functions. Introducing global supervision with CAM besides pixel-wise L1 loss yields improved HSI reconstruction quality. Furthermore, our proposed CAM demonstrates

**Table 3: Ablations on embedding size.**

| Embedding Size | Embedding + Decoder | PSNR |
|----------------|---------------------|------|
| $12 \times 6 \times 16$ | 0.12 + 0.37 MB | 42.16 dB |
| $6 \times 3 \times 32$ | 0.06 + 0.43 MB | 42.58 dB |
| $6 \times 3 \times 16$ | **0.03 + 0.47 MB** | **43.03 dB** |
| $6 \times 3 \times 8$ | 0.02 + 0.48 MB | 42.82 dB |
| $4 \times 2 \times 16$ | 0.02 + 0.48 MB | 42.6 dB |

superior performance compared to the commonly used SSIM, indicating that CAM is more suitable for HSI reconstruction.

**Table 4: Ablation on the loss function.**

| | 0.5M | 1M | 2M |
|---|------|-----|-----|
| L1 | 43.78 | 44.66 | 45.97 |
| L1+SSIM | 43.75 | 44.82 | 46.26 |
| L1+CAM | **44.14** | **45.55** | **47.85** |

**Classification.** Table 5 illustrates the gradual transformation from using the original SF to our proposed method dedicated for compressed HSI. By adjusting the compressed reconstruction to adapt to classification task through ASW, and improving the generalization of classification model with augmented compressed HSI samples through ISI, we can achieve considerable accuracy improvement under high compression ratio.

**Table 5: Ablation on proposed classification.**

| | ASW | ISI | OA (%) | AA (%) | $\kappa$ |
|---|-----|-----|--------|--------|----------|
| case 1 (SF) | ✗ | ✗ | 79.15 | 84.27 | 0.7633 |
| case 2 | ✓ | ✗ | 84.75 | 89.88 | 0.8268 |
| case 3 (Ours) | ✓ | ✓ | **87.03** | **90.99** | **0.8519** |

## 5 CONCLUSION

In this paper, we propose HINER, a novel neural representation for HSI. By explicitly embedding spectral wavelengths and introducing global CAM supervision, HINER effectively exploits correlation within and across spectral bands in HSI. Simultaneously, we thoroughly consider the degradation of downstream classification task induced by lossy compression. Through using ASW for classification-oriented reconstruction and ISI for data augmentation in training classification model, we effectively mitigate the task degradation. Experimental results demonstrate notable improvements in compression efficiency, model convergence, and classification accuracy compared to previous explorations. However, there is still plenty of room to improve our methodology. For instance, it falls short of the latest VVC in certain datasets, and still requires a few labels to supervise the classification model. These encourage us to pursue a more compact representation of HSIs and explore strategies involving soft label supervision in future work.

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
