# OpenReview forum: "HINER: Neural Representation for Hyperspectral Image"
_acmmm.org/ACMMM/2024/Conference — MM2024 Poster_

### Official Review · Reviewer_C7R3 · 2024-05-16

**Rating:** 2
**Confidence:** 3

**Summary:**

The authors propose the HINER for HSI compression and verify the effectiveness of the proposed method through the downstream classification task.

**Strengths:**

Using NR for HSI compression is interesting and rare.

**Limitations:**

1) The main contributions of the manuscript are unclear. The author claims that the main contribution is HINER for HSI compression. However, from the manuscript, it seems that ASW and ISI for classification are also the main contributions (L175). It seems that these modules are not related to the title of the article.

2) From the perspective of HSI compression, the novelty of the model structure and CAM is limited.

3) The author needs to choose more SOTA HSI compression methods for comparison. Besides, in addition to the PSNR and bpppb, values of other compression metrics need to be provided.

4) From the perspective of HSI classification, the novelty of ASW and ISI is also limited. For example, it seems that ASW is very similar to spectral attention.

5) The readability of the article needs to be further improved.

**Suitability:**

2

---

### Official Review · Reviewer_fNYa · 2024-05-18

**Rating:** 4
**Confidence:** 3

**Summary:**

The paper introduces HINER, a novel approach for compressing hyperspectral images (HSI) using Neural Representation. HINER achieves high compression performance while maintaining accuracy for downstream classification tasks.

**Strengths:**

Drawing inspiration from video processing worlk (Hnerv)[1] and designing the network based on the characteristics of HSI is a sensible approach.

The representation is clear and easy to follow.

The experimental analysis is thorough, and the results are impressive.

[1] Hao Chen, Matthew Gwilliam, Ser-Nam Lim, and Abhinav Shrivastava. Hnerv: A hybrid neural representation for videos. In Proceedings of the IEEE/CVF Conference on Computer Vision and Pattern Recognition, pages 10270–10279, 2023.

**Limitations:**

The analysis of the differences between video and HSI has been mentioned in [2], please kindly cite this work.

Spectral angle mapper measures the similarity between spectrums by treating spectrum at each pixel as a high-dimensional vector and calculating the angle between vectors.  However, the Content Angle Mapperr calculates the spatial vectors of two bands, which seems inconsistent with the motivation. Please further explain the rationale behind the motivation.


[2] Xiao J, Ji Y, Wei X. Hyperspectral Image Denoising with Spectrum Alignment[C]//Proceedings of the 31st ACM International Conference on Multimedia. 2023: 5495-5503.

**Suitability:**

2

---

### Official Review · Reviewer_gEso · 2024-06-01

**Rating:** 3
**Confidence:** 3

**Summary:**

The main content of this paper is the proposal of HINER, a neural representation framework for hyperspectral image (HSI) compression. The authors address the challenge of efficiently compressing HSIs, which contain a large number of spectral bands, by leveraging spectral redundancy and capturing pixel correlations. They introduce HINER, which incorporates wavelength embeddings and fully supervises reconstruction fidelity within each spectral band. The authors also investigate the issue of accuracy degradation in downstream classification tasks on compressed HSIs and propose solutions to mitigate this problem.

**Strengths:**

1. Novelty: HINER introduces explicit encoding of wavelengths and global CAM loss to effectively exploit spectral redundancy in HSI samples. This explicit embedding allows for better characterization and exploitation of cross-band correlation.
2. Theoretical approach and technical correctness: HINER utilizes a learnable encoder and decoder to generate spectral embeddings and reconstruct the HSI. The encoder is designed with low computational complexity and efficient spectral correlation capturing. The optimization objective of HINER is formulated to minimize distortion between the input and reconstructed HSI.

**Limitations:**

One potential weakness could be the lack of novelty in the proposed HINER (Hyperspectral Image Neural Representation) method. The paper mentions that over the past few years, there has been a growing interest in leveraging deep learning techniques for HSI compression, and previous works have already explored neural representation for HSI compression. This suggests that the HINER method may not introduce significant new ideas or approaches compared to existing methods.
Another potential weakness could be the limited evaluation of the proposed method. While the paper mentions experimental results demonstrating superior compression efficiency and comparable computational complexity, it does not provide detailed information on the evaluation methodology or results. It is important for a research paper to have a thorough evaluation to validate the effectiveness and performance of the proposed method.

**Suitability:**

2

---

### Official Review · Reviewer_ZPL2 · 2024-06-02

**Rating:** 4
**Confidence:** 3

**Summary:**

This paper proposes HINER, a novel approach for compressing HSI using Neural Representation. The proposed framework fully exploits inter-spectral correlations by explicitly encoding of spectral wavelengths and achieves a compact representation of the input HSI sample through joint optimization with a learnable decoder.

**Strengths:**

This paper is well-written, and the experiments are sufficient.

**Limitations:**

1. Regarding the encoding and decoding time, VVC or HEVC (video coding standards) use CPU compression, while the proposed model uses GPU, which is unfair. How effective is CPU testing upon the proposed method?
2. For the proposed coding method, is the training time corresponds to the encoding time?
3. For different compression rates, they can be controlled by the parameter quantity of the network or by the quantization configuration of the network. How to balance the two?
4. Do we need an additional network structure for different resolutions? In the other words, each type resolution need a different network structure.
5. Does the generated embedding need to be quantified? If so, how does the quantification of embedding balance with model compression?

**Suitability:**

3

---

### Official Review · Reviewer_W5rq · 2024-06-08

**Rating:** 3
**Confidence:** 2

**Summary:**

The paper introduces HINER, a new neural representation method for hyperspectral image compression, effectively leveraging spectral wavelength embedding and adaptive spectral weighting to enhance compression efficiency and downstream classification performance.

**Strengths:**

HINER demonstrates superior compression performance and task accuracy across various hyperspectral image datasets, with a lightweight and computationally efficient design that addresses both global and local spectral information.

**Limitations:**

1.The paper claims that HINER is computationally efficient. Can the authors provide a more detailed analysis of the computational complexity, including memory usage and runtime, compared to the baseline methods?
2.How does the explicit spectral wavelength embedding improve the performance of HINER compared to traditional neural representations? Can the authors provide more detailed examples or visualizations?
3.The article mentions that HINER embeds spectral correlations by explicitly encoding spectral wavelengths. Can you explain why this embedding method is more effective than the content-independent NeRV? In addition, can you explore the impact of different wavelength encoding methods on model performance?
4.The article mentions that HINER's performance on some datasets is not as good as VVC. Can you analyze the reasons for this difference? In addition, can you discuss how to improve HINER to narrow the performance gap with VVC?

**Suitability:**

2

---

### Meta-Review · Area_Chair_pPdq · 2024-06-27

**Recommendation:** Accept (Poster)
**Confidence:** 4

**Metareview:**

1 Weak Accept, 2 Borderline Accept, 2 Borderline Reject

Three reviewers tend to accept this paper and two reviewers recommends "Borderline Reject". I agree with the majority side. Please address the following issues in the final paper.

-- lack of novelty in the proposed HINER
-- unclear contributions
-- How the explicit spectral wavelength embedding improves the performance of HINER
-- why this embedding method is more effective than the content-independent NeRV
-- how to improve HINER to narrow the performance gap with VVC
-- How to balance the parameter quantity and quantization configuration of the network
-- how the quantification of embedding balances with model compression
-- more experiments and comparison